# Characterization of Films Produced with Cross-Linked Cassava Starch and Emulsions of Watermelon Seed Oils

**DOI:** 10.3390/foods11233803

**Published:** 2022-11-25

**Authors:** Julio Colivet, Vitor Augusto dos Santos Garcia, Rodrigo Vinicius Lourenço, Cristiana Maria Pedroso Yoshida, Alessandra Lopes de Oliveira, Fernanda Maria Vanin, Rosemary Aparecida de Carvalho

**Affiliations:** 1Department of Food Engineering, Faculty of Animal Science and Food Engineering, University of São Paulo—USP, Av. Duque de Caxias Norte 225, Pirassununga 13635-900, SP, Brazil; 2Technology Department, State University of Maringá, Av. Ângelo Moreira da Fonseca 1800, Umuarama 87506-370, PR, Brazil; 3Institute of Environmental, Chemical and Pharmaceutical Science, UNIFESP—Federal São Paulo University, Rua São Nicolau 210, Diadema 09913-030, SP, Brazil

**Keywords:** biodegradable films, bioactive compounds, emulsions, contact angle kinetics

## Abstract

Starches are promising molecules in the production of edible films. However, the hydrophilic nature of these materials is among the main limitations of packaging based on natural polymers. An underexplored alternative is the incorporation of emulsions. This work aimed to produce films based on crosslinked cassava starch with emulsions based on watermelon seed oil (WSO) extracted with pressurized ethanol. The effect of incorporating watermelon seed oil emulsion (WSOE) on the microscopic, structural, mechanical, hydrophilic, and thermal properties of films was analyzed. The internal structure and roughness of the films were significantly affected by increasing WSOE concentration. The WSOE incorporation increased the elongation capacity of the films and reduced the strain at break. WSOE concentrations did not significantly affect the water solubility, permeability, and X-ray diffraction but decreased the wettability of the films. The analysis of the thermal properties showed that the films did not present phase separation in the studied temperature range. Overall, WSOE improved the properties of the films based on cross-linked cassava starch, but it is necessary to optimize the production conditions of the films. These materials may potentially be used as biodegradable food packaging, controlled-release films, and edible coatings in food protection.

## 1. Introduction

Starches are highly versatile macromolecules in the production of biodegradable films. In addition to the native form, modified starches are industrially produced and applied in different areas. Some of the modifications applied are physical treatments (shear force, blending, and thermal treatment) and chemical modifications (crosslinking, acetylation, oxidation, and application of two modifications, generally combinations of crosslinking and acetylation) [1].

Modified starch films have been reported in the literature. Acetylation treatments on rice starches produced homogeneous films with an elongation at break higher than native starches films [2]. However, the films preserved their hydrophilic characteristics. Chemically modified cassava starch using potassium periodate to produce biodegradable films was characterized by a higher tensile strain at break and a reduced water solubility than the films without modification [3]. Double modification (debranched/hydroxypropylated) in corn starch increased the elongation and the tensile strength at break of films; however, the oxygen and water vapor barrier were inferior to films produced with native or modified starches [4].

The modifications applied to the starches are alternatives to improve the films properties compared to materials produced from native starches. In many cases using only modified starches, the films preserved their hydrophilic characteristics that did not improve the barrier and mechanical properties [5]. The incorporation of hydrophobic substances has been explored. Shi et al. [6] reported that combining polysaccharides or proteins with lipids to produce films is a promising alternative to improve the barrier properties of biopolymer-based films. Films produced with starch and fatty acids formed homogeneous microstructure matrices [7]. The glycerol monostearate (plasticizer) addition in yam starch polymeric matrices caused slight phase separations after drying, which is associated with the hydrophobicity characteristic of the monoglyceride [8].

Incorporating lipids in the formulation of biodegradable matrices represents a technical challenge, as they are generally incompatible with natural polymers due to their hydrophilic characteristics. The literature has reported techniques that include different ways to incorporate lipids into polymeric matrices, such as an additional layer on a formed film [9], application inside layers of films (laminates) [10], or in the form of emulsions [11].

Phase separation was observed in cassava starch films containing different levels of stearic acid concentration, verifying that the stearic acid affected the mechanical properties, increased the water vapor permeability (WVP) properties, and reduced water solubility [12]. Depending on the concentration of the fatty acid incorporated into corn starch films, an additional barrier to water vapor and an improvement in the mechanical properties t were observed, with 4 g of stearic acid and 24 g of glycerol/100 g starch dry as the ideal concentration [7]. Muscat et al. [13] evaluated the film formation and hydrophobicity of high amylose starch and natural waxes (bee, candelilla, and carnauba) in the presence and absence of Tween-80. They reported that the lipid’s presence increased the hydrophobicity and, in some treatments, caused shrinkage and discontinuity in the structure of the polymer matrix. The incorporation of surfactants (tween 20, soy lecithin, and spam) in potato starch films reduced WVP and negatively affected the mechanical properties (break strength and elongation) [14].

A little-explored alternative is the incorporation of previously stabilized emulsions in filmogenic solutions. The production of emulsions using the Pickering (PE) technique is a methodology that stabilizes the oil–water interface with solid particles through an absorption system [15]. Incorporating PE based on paraffin oil and chitosan nanocrystals in starch-based matrices formed highly stable and homogenous films, improving the mechanical strength [16]. The WVP of potato starch films reduced when the PE of olive oil and zein particles in film-forming solutions [17] were incorporated. Chitosan films containing PE from corn oil and zein particles presented reduced oxygen and water vapor permeabilities [6].

Essential oils [18,19], phenolic compounds [20], and vegetable oils [17] are functional compounds used in the production of PE. Other vegetable oils with functional properties are the seed oils of fruits, such as watermelon [21]. Watermelon seed oil is an excellent source of linoleic acid (62–68%) and oleic acid (13–15%) [22]. Acar et al. [23] determined that watermelon seed oil has phenolic compounds with antioxidant capacity and is an important source of tocopherols.

In this context, the present study evaluated the effect of incorporating Pickering-type emulsions of WSO extracted with pressurized liquids on the properties of modified cassava starch films.

## 2. Materials and Methods

### 2.1. Materials

Ruby Watermelons were purchased from a local market in Pirassununga (São Paulo, Brazil). The emulsions were produced using Octenyl succinic anhydride (OSA)-modified maize starch (OSA, OSA EP 1002, Cargill, São Paulo, Brazil). Film production was performed using T13-270 cross-linked starch donated by Gemacom Tech S.A. (Sao Paulo, Brazil). The plasticizer sorbitol was purchased from Vetec Fine Chemicals Ltd., São Paulo, Brazil. Extractions were performed with absolute ethyl alcohol (Synth, São Paulo, Brazil). Total phenolics were determined using the Folin–Ciocalteu reagent (Sigma-Aldrich, St. Louis, MO, USA) and gallic acid (Sigma-Aldrich, USA). The analysis of antioxidant activity was performed using the reagent 2,2-diphenyl-1-picrylhydrazine (Sigma Aldrich, USA).

### 2.2. Watermelon Seed Oil (WSO) Extraction

The WSO was extracted according to the methodology described by Colivet et al. [24]. The pressurized liquid extraction (PLE) technique (ASE 150 equipment, Dionex, Sunnyvale, CA, USA) was used. The dehydrated and ground watermelon seeds (18 mesh), absolute ethanol at 60 °C, and a cell of 100 mL (constant mass/solvent ratio of 0.30 g/mL) were used. Five extraction cycles were performed (6 min intervals) in an intermittent process. The products of each extraction cycle were homogenized, and the solvent was evaporated at 40 °C using a rotary evaporator (IKA RV 05 IKA, IKA-Werke, Staufen, Germany).

### 2.3. Film Production

The films were produced with the casting technique. The plasticizer was solubilized in distilled water (30 g of sorbitol/100 g of starch), then the starch (4 g/100 g of solution) was dispersed under magnetic stirring for 30 min at room temperature. The dispersion was heated at 90 °C for approximately 10 min under stirring (IKA^®^ C-MAG HS 7, Staufen, Germany). After this period, the solution was cooled (room temperature) to 60 °C, and different concentrations of watermelon seed oil emulsion (C_WSOE_; 0, 0.1, 0.2, 0.3, 0.4, and 0.5 g of oil/100 g) were added. Watermelon seed oil emulsions were produced according to Dokić et al. [25], with modifications. OSA starch (15 g) was dispersed in 100 mL of distilled water, and the dispersion was kept at 50 °C for 20 min (IKA^®^ C-MAG HS 7 shaker, Staufen, Germany). Subsequently, watermelon seed oil (12 g) was incorporated into the starch solution, and the homogenization of the system was carried out at 9500 rpm (20 min, 25 ± 2 °C) using an UltraTurrax stirrer (IKA, T25, Staufen, Germany). The filmogenic solutions with the added emulsions were homogenized at 5000 rpm for 5 min, using an UltraTurrax shaker (IKA, T25, Germany). Afterward, the solution was applied to acrylic plates (12 × 12 cm) and dried (30° C, 24 h) in a convection oven (Marconi, MA 037, São Paulo, Brazil). The thickness of the matrices (arithmetic mean of ten random measurements over the area of the film) was analyzed using a digital micrometer (Mitutoyo, Tokyo, Japan).

Film analyses were performed after conditioning the matrices at 58% RH (NaBr saturated saline solution, 25 ± 2 °C) for 5 days. In the case of microscopy (scanning electron and atomic force) and X-ray diffraction, the films were conditioned in desiccators with silica gel for 7 days (25 ± 2 °C).

### 2.4. Characterization of Films

#### 2.4.1. Scanning Electron Microscopy (SEM)

The film structures (surface and internal) were analyzed using a scanning electron microscope (Tabletop Hitachi TM3000, Hitachi Hight-Technology Corporation, Tokyo, Japan) at 5 kV. The films were immersed and fractured in liquid nitrogen to analyze the internal microstructure.

#### 2.4.2. Atomic Force Microscopy

Surface roughness analysis of films with and without the incorporation of watermelon seed oil emulsion (WSOE) was performed using an atomic force microscope (AFM, Solver Next, NT-MDT, Moscow, Russia) in semi-contact mode. The images were analyzed using the Image Analysis 3.1.0.0 program. The mean square roughness (Rq) and mean roughness (Ra) were calculated using the data plane height deviations (Zj), according to ASME B46.1 methodology [26].

#### 2.4.3. Contact Angle Variation Kinetics (θ)

The θ measurement of the films was performed using an optical tensiometer Attension Theta Lite (Biolin Scientific AB, Gothenburg, Sweden) according to the methodology proposed by Karbowiak et al. [27]. A total of 5 μL of deionized water was placed on the surface of the matrices and the θ was evaluated for 300 s. A sheet of aluminum foil (considered impermeable to water and aqueous solutions) was used to evaluate the evaporation effect, as proposed by Kurek et al. [28]. The contact angle variation kinetics were evaluated using the model by Farris et al. [29] (Equation (1)).
(1)$$\theta \left(t\right)=\theta_{0}exp\left(-kt^{n}\right)$$
where $\theta_{0}=$ the initial contact angle (degrees), k= velocity rate of θ (°s^−1^), *t* = time (s) after the droplet deposition on the surface, *n* = exponent related to the kinetic phenomenon (absorption or spreading) [29].

#### 2.4.4. Moisture Content

The samples were submitted to drying at 60 ± 1 °C until constant mass to determine the water content [30]. The moisture content was used to calculate the (MC) Equation (2):(2)$$MC=\frac{X_{0}-X_{f}}{X_{0}}\times 100$$
where *MC* = moisture content (g of H_2_O/100 g of film),$\;X_{0}=$ $\mathrm{initial}\;\mathrm{mass}\;\mathrm{of}\;\mathrm{the}\;\mathrm{film},\;X_{f}=$ final mass after the drying process.

#### 2.4.5. Water Vapor Permeability (WVP)

The WVP of films with and without watermelon seed oil emulsion (WSOE) was determined using the modified ASTM 96 method [31]. Silica gel was used as a desiccant agent. The permeability cells were stored in desiccators at 75% RH (controlled RH with a saturated saline solution of sodium chloride), and the desiccators were kept at 25 ± 3 °C (BOD Marconi, MA415, Piracicaba, Brazil). The mass gain was determined using an analytical balance (Ohaus Adventurer Pro AV 4101, Ohaus^®^ AV 4101, Parsippany, NJ, USA) for 5 days in 24 h periods. WVP was calculated according to Equation (3).
(3)$$WVP=\frac{∆Wx}{tAP_{0}\left(RH_{1}-RH_{2}\right)}$$
where $\frac{∆W}{t}$ = angular coefficient of the line (mass gain vs time, g/h); *x* = thickness (mm); *A* = area (32.15 cm^2^); *P*_0_ = vapor pressure of pure water at 25 °C (kPa); $\left(RH_{1}-RH_{2}\right)\;$ = relative humidity gradient used in the experiments. At 25 °C, *P* = 3.169 kPa [32].

#### 2.4.6. Mechanical Properties

Elongation (E), elastic modulus (EM), and stress at break (RT) were determined using the ASTM D882-02 method [33]. Tests were performed using a TA-TX plus texturometer (Stable Micro Systems, Surrey, UK). Samples measuring 12 × 2.5 cm were used to conduct the analyses, with velocity and initial distance set at 1.0 mm/s and 100 mm, respectively. Mechanical properties were determined using Texture Expert software (Stable Micro Systems, Surrey, UK).

#### 2.4.7. Thermal Properties

The glass transition temperatures ($Tg_{1}$ and $Tg_{2}$), melting temperature ($T_{m}$), and enthalpy ($∆H_{g}$) were determined according to Sobral and Habitante [34], using a TA2010 differential scanning calorimeter (TA instruments, New Castle, DE, USA). Samples (10 mg) were weighed on an analytical balance (±0.00001) and placed in a desiccator containing NaBr saturated saline solution (58% RH, 25 ± 3 °C) for 2 weeks. Temperature ranges from −20° to 180 °C were used (heating ramp of 10 °C/min). Two scans were performed, and parameters were calculated using Universal Analysis V1.7 software (TA instruments).

#### 2.4.8. X-ray Diffraction

X-ray diffraction analyses were performed using a Rigaku Miniflex 600 diffractometer (Rigaku Co., Tokyo, Japan), operating at 30 kV and 20 mA using CuKa radiation of 1.5406 Å as the X-ray source. The analysis was performed by changing 2θ in the region between 2 and 60 A (scan speed of 5°/min, room temperature).

#### 2.4.9. Color Parameters

Chroma parameters *a**, *b**, and luminosity (*L**) were analyzed using a HunterLab colorimeter (Miniscan XE plus, Reston, VA, USA) according to the method reported by Gennadios et al. [35]. Colorimeter calibration was performed using standardized black and white plates. The parameters *a**, *b**, and *L**were determined by placing the films on the white plate (color parameters, *L** = 93.9: *a** = −0.8 and *b** = 1.2). The total color difference was determined according to Equation (4).
(4)$$∆E=\sqrt{{\left(L_{p}{}^{*}-L^{*}\right)}^{2}+{\left(a_{p}{}^{*}-a^{*}\right)}^{2}+{\left(b_{p}{}^{*}-b^{*}\right)}^{2}}$$
where *L_p_**, *a_p_**, and *b_p_** are the values of chroma *a**, chroma *b** and luminosity of the of the film without the incorporation of WSOE (control film), and *a**, *b*, and *L** are the values obtained for films with the incorporation of WSOE.

#### 2.4.10. Total Phenolic Content

The total phenolic concentration of the films was determined using the method described by Singleton et al. [36]. The extraction of the film samples was carried out by dispersing 0.10 g of the matrices in distilled water (1 mL) at 25 °C [37]. The dispersion was centrifuged (4000 rpm, 10 min) at 25 ± 2 °C, and the total phenolic content was determined according to Singleton et al. [36]. The supernatant (0.5 mL) was added to 2 mL of Folin–Ciocalteau reagent previously diluted in distilled water (1:5 H_2_O). The solution rested for 3 min and then mixed with 2.5 mL of Na_2_CO_3_ (20%, *w*/*v*) using a vortex (IKA vortex 1V1, Staufen, Germany). The solution was kept in the dark for 2 h, and the absorbance was determined at 760 nm using a spectrophotometer (Biospectro, SP-22, São Paulo, Brazil). The calibration curve was prepared using gallic acid as an external standard in the concentration range of 0 and 80 µg of gallic acid/mL. The content of total phenolics was expressed as mg of gallic acid equivalents/g of dry film.

#### 2.4.11. Antioxidant Activity (AA)

AA was calculated using the *DPPH*• (2,2-diphenyl-1-picrylhydrazyl) method, according to Kowalczyk and Biendl [37]. The films (0.1 g) were dispersed in 5 mL of distilled water, then kept at rest for 30 min and centrifuged at 6000× *g* at 25 °C for 10 min (Eppendorf Centrifugue 5415C, Vaudaux, Schonenbuch, Switzerland). A total of 1 mL of the obtained solution was added to 3 mL of *DPPH*• solution (0.1 mM in methanolic solution). The solution was kept at rest for 60 min in the dark at 25 ± 3 °C. The solution absorbance was determined at 517 nm using a spectrophotometer (Biospectro, SP-22, São Paulo, Brazil). *AA* was determined according to Equation (5).
(5)$$AA\left(\%\right)=\left(\frac{1-As}{A_{DPPH}}\right)$$
where: *AA* is the antioxidant activity, As is the absorbance of the *DPPH*• solution containing the sample aliquot, and *A_DPPH_* is the absorbance of the *DPPH*• solution.

### 2.5. Statistical Analysis

An analysis of variance (ANOVA) was performed, and the difference between the means was determined using Duncan’s multiple range test (95% confidence). The analysis was performed using the Statistica 10.0 software (Stat Soft Inc., Tulsa, OK, USA). All analyses were performed in triplicate.

## 3. Results and Discussion

### 3.1. SEM

The control film (C_WSOE_ = 0, Figure 1) presented surface irregularities compared to the films with WSOE (Figure 1). The presence of particles was observed on the surface in the films with WSOE, and increasing the C_WSOE_ showed more WSOE particles on the film surface.

The internal microstructure of the control film (C_WSOE_ = 0, Figure 1) revealed a cohesive and compact structure. The films with WSOE showed particle dispersion in the structure, showing dark areas (oil particles). Similar behavior was observed in the internal structure; increasing C_WSOE_ caused the dispersion of oil particles in the internal network. The exudation of plasticizer or oil on the film surfaces was not visually observed, indicating that incorporating the emulsified oil contributed to stabilizing the system. Thus, in this context, the application of emulsions in cross-linked starch polymeric matrices indicates a good alternative in incorporating hydrophobic compounds. It was possible to verify that the particles remained intact, as also observed by Jiménez-Saelices et al. [16].

### 3.2. Atomic Force Microscopy

Roughness means square ($R_{q}$) and means roughness ($R_{a}$) were significantly affected by WSOE concentration (Figure 2, Table 1). There was no correlation between the increase in C_WSOE_ and roughness. As shown in Figure 1, the results are possibly related to the heterogeneous distribution of particles in the polymer matrix. Overall, the incorporation of WSOE increased the roughness of the films, which is a result related to the distribution of emulsions in the film structure. The roughness is an important parameter, as it affects properties of the films such as appearance, surface properties, and possible interactions with other substances (water and some food components).

Higher roughness values were observed in films based on Konjac glucomannan with different concentrations of sunflower oil PE stabilized with cellulose nanofibers and soy protein isolates, compared to the films without added PE [38]. However, the roughness decreased when incorporating PE with oil phase concentrations equal to 70% [38].

### 3.3. Contact Angle

Incorporating WSOE in the films reduced the initial contact angle (*θ*_0_, Table 2). In all cases, $\theta_{0}$ was lower than the value obtained in films produced without WSOE (C_WSOE_ = 0, Table 2).

The behavior of the contact angle may be related to the composition, temperature, and structural properties, such as the roughness of the materials [39]. In this study, the incorporation of WSOE did not indicate a relation between the roughness with the emulsion concentration and the initial contact angle. However, the treatment with the highest emulsion concentration had a lower $\theta_{0}$ and a higher roughness value. The results are possibly associated with particle size and distribution of the polymer matrix, in addition to the composition of watermelon seed oil. Liu et al. [38] reported that adding sunflower essential oil emulsions increased the contact angle in Konjac glucomannan-based films.

Farris et al. [29] proposed that the kinetics of the contact angle can be measured using a semi-empirical exponential equation, where “*k*” is related to the evolution of the contact angle velocity. This study showed a reduction in the speed of variation of the contact angle “*k*”, increasing the WSOE concentration. The decrease in “*k*” may be related to the presence of emulsion particles on the surface of the films.

Wiacek and Dul [40] reported higher “*k*” values (~0.02° s^−1^) for rice starch films modified with phosphatidylcholine molecules. Values greater than 0.1° s^−1^ were obtained in chemically modified starch films [41]. The differences in the values reported in the literature may be associated with the lipids’ presence in the structure and the degree of homogeneity of the matrices.

Farris et al. [29] suggested that the values of “*n*” could be used to determine the mechanisms involved during the deposition of a drop of water on the surface of the film; values of *n* = 0 are associated with the occurrence of absorption phenomena, and values of *n* = 1 are associated with spreading phenomena. In the present study, the parameter “*n*” values were higher than 1, indicating greater spreading than absorption. Incorporating WSOE, the “*n*” values progressively increased with the oil concentration, except for treatments with 0.1–0.3% of oil. According to the results shown in Table 2, the tendency of the films was an increase in spreading rates and a decrease in absorption as the oil concentration increased. Additional studies must be carried out to confirm this hypothesis. There were “*n*” values higher than those found in C_WSOE_ films (~1.2), indicating that the incorporation of nonpolar compounds affects the wettability and surface properties of the films. Gelatin-based films with a hydrophobic plasticizer (acetyl tributyl citrate) presented “*n*” values equal to 0.79. This behavior is associated with the tendency to pure spreading, probably due to the characteristics of the hydrophobic plasticizer [42].

### 3.4. WVP

Water vapor permeability was not significantly affected by the incorporation of watermelon oil emulsions (Table 3), and no significant differences were observed between the films with and without WSOE incorporation. The results may be related to the distribution of oil particles in the polymeric matrix and to the hydrophilic nature of the starch used to produce the emulsion. According to Roy and Rhim [43], incorporating Pickering emulsions (PE) based on clove essential oil stabilized with cellulose fiber in gelatin films reduced WVP. The authors reported that this behavior could be associated with hydroxyl groups in the matrices, which can cause affinity with water molecules. Souza et al. [19] reported an increase in WVP in starch films added with the PE of essential oils and stabilized with nano cellulose, which is justified by the hydrophilic nature of the formulation components.

### 3.5. Mechanical Properties

The incorporation of WSOE significantly reduced the breaking stress (Table 3) compared to the control film (C_WSOE_ = 0), possibly related to an increase in the mobility of the amylose and amylopectin chains with the incorporation of emulsions. Increasing the WSOE concentration did not significantly affect the stress at break, except for the 0.4% treatment. This may be associated with the characteristics of the physical particles and the dimensions in the internal structure of the films. Jiménez-Saelices et al. [16] reported that the PE incorporation reduced the stress at the break of starch-based polymeric matrices, indicating the discontinuity zones caused by emulsions in the polymeric matrix.

Elastic modulus and elongation at break were significantly affected by incorporating WSOE, in which there was an increase in elongation compared to the control films and a reduction in elastic modulus (Table 3). Similar to what was observed for the tensile strength, it was not possible to establish a correlation with emulsion concentration.

The elastic modulus and elongation are related to the mobility of the polymeric chains and the chemical composition of the polymer. Liu et al. [39] found that the PE incorporation of sunflower oil into Konjac glucomannan films caused an increase in elongation when the emulsion concentration was increased. The results suggest that the particles’ concentration, distribution, and size can affect the films’ properties.

### 3.6. X-ray Diffraction

Films with WSOE showed similar characteristics to films without emulsion incorporation (Figure 3). There were crystalline peaks similar to those found in modified cassava starches at 2θ = 20°. Peaks at 2θ = 24.58° were observed for all formulations. According to the diffractograms, the WSOE did not significantly affect the crystallinity of the films produced.

Different studies have reported that incorporating emulsions in starch films promotes the formation of amorphous zones through the disintegration of the starch grain, altering the organization of the amylose and amylopectin chains and increasing the formation of amorphous zones [19,44]. 

### 3.7. Thermal Properties of Films Made from Cross-Linked Cassava Starch and Watermelon Seed Oils

All films presented a melting peak and glass transition temperature in the first scan ($T_{m}\;\mathrm{and}\;Tg_{1}$, respectively) and a Tg in the second scan ($Tg_{2}$). The results indicated no phase separation, regardless of the formulation (Table 4).

The films with WSOE had a lower $T_{g}$ than the film without WSOE. The presence of only one *T_g_* in the first and second scans indicated the incorporation of the emulsions in the matrix. According to Ghanbarzadeh and Oromiehi [45] the presence of more than one glass transition temperature in matrices composed of several substances indicates phase separation and incompatibility of the constituents.

The analysis of ($T_{m}$) and $∆H_{g}$ did not present significant differences in the films with emulsions incorporation.

### 3.8. Color Parameters

The incorporation of WSOE in the polymeric matrices significantly affected the color parameters (Table 5). No significant differences were verified between the formulations with oil concentrations of 0.3, 0.4, and 0.5%. The incorporation of emulsions increased the values of *a**, *b**, and ∆*E* and reduced the values of *L**. The pure cross-linked starch-based film showed a higher *L** value and lower *a** and *b** values, reflecting that the starch-based film was colorless. The addition of WSOE slightly increased the yellowing (*b**-value) of the film due to the yellow-orange color of the oil. The film’s total color difference (ΔE) changed significantly with the addition of WSOE. The application of WSOE may affect the film’s ability to absorb and transmit light. Additional studies are needed to validate this hypothesis.

Similar results were reported by Liu et al. [46] in starch and PVA matrices added with the PE of curcumin/triglycerides/OSA starch, resulting in a reduced luminosity (*L**) and increased *a** and *b** values, as the emulsion concentration increased. The results obtained may be associated with the concentration of emulsion particles on the surface of the films.

### 3.9. Total Phenolics and Antioxidant Activity

It was observed that the concentration of total phenolics in the films increased significantly with increasing the emulsion concentration (Table 5). The results are related to the increased concentration of watermelon oil in the matrix, and the production process did not cause their degradation. The antioxidant activity was not verified using the *DPPH*• methodology. The results may be related to the extraction method, or the method used.

Priyadarshi et al. [47] reported that incorporating nanoparticles and bioactive compounds into polymeric matrices reduced the antioxidant capacity of the films, which may be associated with the ability of some compounds to generate reactive oxygen species. The incorporation of PE based on clove essential oil and zinc and copper nanoparticles reduced the antioxidant activity of gelatin and agar matrices [48].

## 4. Conclusions

The incorporation of WSOE in crosslinked starch films changed the surface properties, hydrophilicity, and mechanical and structural properties. The film properties were affected by the concentration of WSOE in the films. The incorporation of WSOE did not affect the barrier properties of the films. Although the concentration of phenolic compounds in the films increased as the concentration of WSOE increased, no antioxidant activity was observed. The incorporation of emulsions in natural polymer film represents an important alternative to obtain matrices with homogeneity and stability.

## Figures and Tables

**Figure 1 foods-11-03803-f001:**
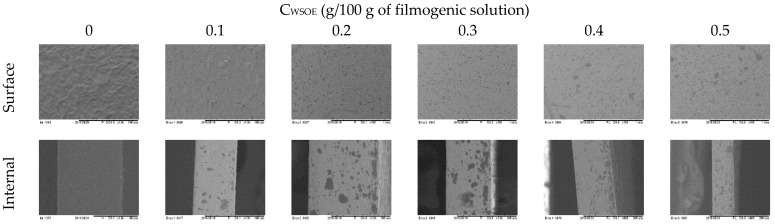
Effect on the surface and internal structure of cross-linked cassava starch-based films with different concentrations of watermelon seed oil emulsions (C_WSOE_, g/100 g of filmogenic solution).

**Figure 2 foods-11-03803-f002:**
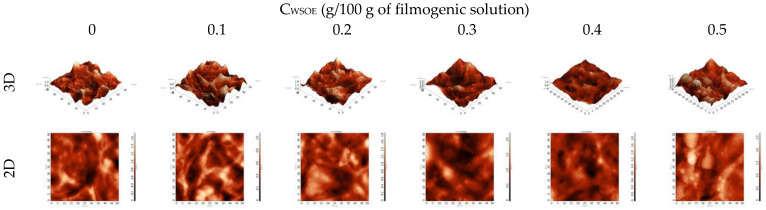
AFM images of films based on cross-linked starch as a function of the incorporation with different concentrations of watermelon seed oil emulsions (C_WSOE_, g/100 g of filmogenic solution).

**Figure 3 foods-11-03803-f003:**
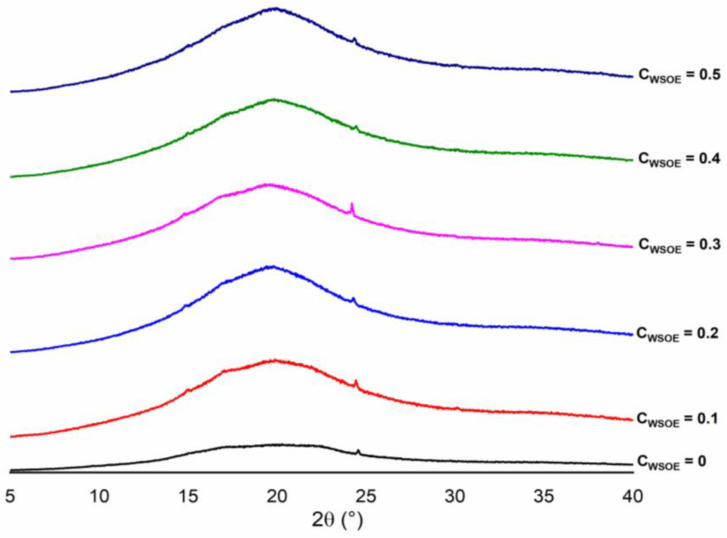
X-ray diffractogram of crosslinked starch-based films with different concentrations of watermelon seed oil emulsions (C_WSOE_, g/100 g of filmogenic solution).

**Table 1 foods-11-03803-t001:** Effect on the average roughness ($R_{a})$ and the roughness mean square ($R_{q}$) of cross-linked cassava starch-based films with different concentrations of watermelon seed oil emulsions (C_WSOE_, g/100 g of film-forming solution).

C_WSOE_	$R_{a}\;(\mathrm{nm})$	$R_{q}\;(\mathrm{nm})$
0	242 ± 33 ^ab^	305 ± 43 ^ab^
0.1	196 ± 19 ^a^	242 ± 23 ^a^
0.2	321 ± 47 ^c^	409 ± 63 ^c^
0.3	301 ± 79 ^bc^	371 ± 89 ^bc^
0.4	261 ± 21 ^b^	324 ± 28 ^b^
0.5	327 ± 32 ^c^	415 ± 43 ^c^

Different lowercase letters in the same column indicate significant differences (*p* < 0.05) by Duncan’s multiple range test.

**Table 2 foods-11-03803-t002:** Effect on the kinetic parameters using the model of Farris et al. [29] of cross-linked cassava starch-based films with different concentrations of watermelon seed oil emulsions (C_WSOE_, g/100 g of film-forming solution).

C_WSOE_	$\theta_{0}$ (°)	n	k (°s^−1^)	R^2^
0	88.50	1.17	1.0 × 10^−3^	0.92
0.1	61.24	1.09	1.8 × 10^−3^	0.96
0.2	62.37	0.79	8.3 × 10^−3^	0.97
0.3	60.28	1.21	1.1 × 10^−3^	0.97
0.4	62.06	1.48	6.5 × 10^−5^	0.97
0.5	47.17	1.62	7.4 × 10^−5^	0.84

**Table 3 foods-11-03803-t003:** Effect on moisture content (MC), tensile strength (TS), elongation (E), elastic modulus (EM), and water vapor permeability (WVP) of cross-linked cassava starch-based films with different concentrations of watermelon seed oil emulsions (C_WSOE_, g/100 g of filmogenic solution).

C_WSOE_	MC (g of H_2_O/100 g of Film)	TS (MPa)	E (%)	EM (MPa)	WVP (g mm/h m^2^ kPa)
0	8.7 ± 0.3 ^a^	9.5 ± 0.5 ^d^	19.8 ± 2.2 ^ab^	589.5 ± 66 ^d^	0.18 ± 0.01 ^a^
0.1	12.9 ± 0.1 ^a^	6.7 ± 1.2 ^c^	25.9 ± 6.9 ^b^	414.9 ± 58 ^c^	0.20 ± 0.01 ^b^
0.2	12.3 ± 0.4 ^b^	6.4 ± 1.0 ^bc^	13.4 ± 8.1 ^a^	375.2 ± 59 ^c^	0.21 ± 0.02 ^b^
0.3	12.6 ± 0.1 ^bc^	6.7 ± 1.0 ^c^	13.3 ± 3.2 ^a^	373.7 ± 97 ^c^	0.19 ± 0.01 ^ab^
0.4	12.5 ± 0.2 ^bc^	4.2 ± 1.0 ^a^	48.5 ± 13.8 ^d^	192.4 ± 82 ^a^	0.18 ± 0.01 ^a^
0.5	12.5 ± 0.1 ^bc^	5.6 ± 1.5 ^b^	39.1 ± 15.8 ^c^	291.4 ± 125 ^b^	0.18 ± 0.01 ^a^

Different lowercase letters in the same column indicate significant differences (*p* < 0.05) by Duncan’s multiple range test.

**Table 4 foods-11-03803-t004:** Effect on glass transition temperature (*T_g_*), melting temperature (*T_m_*) and enthalpy $\left(∆H_{g}\right)$, during the first and second scans of cross-linked cassava starch-based films with different concentrations of watermelon seed oil emulsions (C_WSOE_, g/100 g of film-forming solution).

	First Scan			Second Scan
C_WSOE_	$Tg_{1}$	$T_{m}$	$∆H_{g}$	$Tg_{2}$
0	−28.77 ± 0.14 ^c^	53.43 ± 5.04 ^a^	3.26 ± 0.26 ^b^	−28.17 ± 1.38 ^a^
0.1	−30.26 ± 0.92 ^bc^	49.59 ± 1.73 ^a^	1.38 ± 0.18 ^a^	−21.19 ± 8.34 ^b^
0.2	−31.89 ± 2.08 ^ab^	49.15 ± 2.95 ^a^	2.02 ± 0.92 ^a^	−31.71 ± 1.47 ^a^
0.3	−31.44 ± 0.60 ^ab^	50.00 ± 5.11 ^a^	1.54 ± 0.09 ^a^	−31.79 ± 0.71 ^a^
0.4	−31.75 ± 0.54 ^ab^	49.09 ± 0.65 ^a^	2.34 ± 0.89 ^ab^	−32.54 ± 0.87 ^a^
0.5	−32.04 ± 0.16 ^a^	53.08 ± 6.75 ^a^	2.42 ± 0.89 ^ab^	−32.40 ± 0.46 ^a^

Different lowercase letters in the same column indicate significant differences (*p* < 0.05) by the Duncan’s multiple range test.

**Table 5 foods-11-03803-t005:** Effect on the luminosity (*L**), chroma *a**, chroma *b**, total color difference, and total phenolic concentration (C_TF_, mg of gallic acid equivalents/100 g of film) of cross-linked cassava starch-based films with different concentrations of watermelon seed oil emulsions (C_WSOE_, g/100 g of film-forming solution).

C_WSOE_	*L**	*a**	*b**	∆E	C_TF_
0	91.14 ± 0.22 ^de^	−1.42 ± 0.01 ^a^	0.17 ± 0.05 ^a^	0 ± 0.0 ^a^	0.01 ± 0.001 ^a^
0.1	90.02 ± 0.31 ^c^	−0.29 ± 0.19 ^b^	7.19 ± 1.02 ^b^	7.20 ± 1.04 ^b^	0.19 ± 0.10 ^b^
0.2	89.56 ± 0.55 ^bc^	0.66 ± 0.49 ^c^	12.08 ± 2.14 ^c^	12.20 ± 2.21 ^c^	1.14 ± 0.20 ^b^
0.3	88.42 ± 1.75 ^a^	1.19 ± 0.49 ^d^	14.60 ± 2.55 ^d^	14.97 ± 2.66 ^d^	3.04 ± 0.59 ^c^
0.4	88.92 ± 0.58 ^ab^	1.29 ± 0.50 ^d^	14.84 ± 1.77 ^d^	15.12 ± 1.71 ^d^	4.51 ± 0.94 ^d^
0.5	87.99 ± 0.6 ^a^	2.25 ± 0.0 ^e^	18.08 ± 1.85 ^e^	18.55 ± 2.02 ^e^	5.68 ± 0.97 ^d^

Different lowercase letters in the same column indicate significant differences (*p* < 0.05) by the Duncan’s multiple range test.

## Data Availability

Not applicable.

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
