# Peer review of "Characterization of Films Produced with Cross-Linked Cassava Starch and Emulsions of Watermelon Seed Oils"

_foods, 2022, doi:10.3390/foods11233803_

Round 1

Reviewer 1 Report

- The proof reading and editing of the English language is required for all the text. There are many errors and mistakes in the text. Extensive editing of the English language and style is needed.

- Title: The title could be shorter. "Extracted with pressurized ethanol" is not necessary in the title.

- Line 109: "at a temperature of 60 °C" what does it mean? Please rewrite this sentence.

- Tables: Table 1: It seems that the statistical analysis of the data wasn't carried out correctly! Please correct the statistical comparison and revise the letter according to Duncan's test. Apply for all the tables.

- This format "authors' names reported" is used many times in the manuscript; please use different styles of writing to enhance the quality of the manuscript.

Author Response

We thank you for the opportunity to improve the writing and presentation of the paper entitled “Characterization of films produced with cross-linked cassava starch and emulsions of watermelon seed oils extracted with pressurized ethanol”.

The comments made by reviewers with their respective corrections are presented below. Most of the reviewers' recommended observations were accepted and completed, including formatting errors and the quality of the images.

We acknowledge in advance all the cooperation provided by you during the job evaluation process, and we hope that our work will be published in your prestigious journal.

01) The proof reading and editing of the English language is required for all the text. There are many errors and mistakes in the text. Extensive editing of the English language and style is needed.

The text was thoroughly revised and the English writing was carried out by a company specialized in reviewing scientific articles.

02) Title: The title could be shorter. "Extracted with pressurized ethanol" is not necessary in the title.

The title has been simplified.

03) Line 109: "at a temperature of 60 °C" what does it mean? Please rewrite this sentence.

The phrase "at a temperature of 60 °C" has been replaced by "at 60 °C".

04) Tables: Table 1: It seems that the statistical analysis of the data wasn't carried out correctly! Please correct the statistical comparison and revise the letter according to Duncan's test. Apply for all the tables.

A review of the tables was performed and the statistical analysis was performed again.

Reviewer 2 Report

Characterization of films produced with cross-linked cassava starch and emulsions of watermelon seed oils extracted with pressurized ethanol by Julio Colive et al, is a good study. The paper is very well written but I have some questions in the methodology.

Even though it’s a very good study and the author did their best, but I would like to ask why the author’s didn’t checked the following parameters in this study?

Viscosity of the watermelon seed oil.

colorimetric and texture study.

Sensory Evaluation of Emulsions

As in Figure 1 they have checked Effect on the surface and internal structure of cross-linked cassava starch-based 273 films with different concentrations of watermelon seed oil emulsions (CWSOE, g/100 g of 274 filmogenic solution) but they didn’t check the stability of systems because the selection of emulsion components, is to subject them to external influences such as centrifugal force during the centrifugation process.

A lot of work has been done on watermelon seed oil and emulsions, what is the novelty of your work? How this is different from other research work done previously.

In conclusion the authors mentioned in the last Although the concentration of phenolic compounds in the films increased as the concentration of WSOE increased, no antioxidant activity was observed. This was probably associated with the composition of the polymeric matrix and the presence of compounds responsible for oxidation.

  Probably? It means the authors are not sure about it.

Author Response

We thank you for the opportunity to improve the writing and presentation of the paper entitled “Characterization of films produced with cross-linked cassava starch and emulsions of watermelon seed oils extracted with pressurized ethanol”.

The comments made by reviewers with their respective corrections are presented below. Most of the reviewers' recommended observations were accepted and completed, including formatting errors and the quality of the images.

We acknowledge in advance all the cooperation provided by you during the job evaluation process, and we hope that our work will be published in your prestigious journal.

01)Even though it’s a very good study and the author did their best, but I would like to ask why the author’s didn’t checked the following parameters in this study? Viscosity of the watermelon seed oil. Colorimetric and texture study. Sensory Evaluation of Emulsions.

Response: The main objective of the work is incorporating hydrophobic compounds in films based on natural polymers, aiming at improving the barrier properties and providing additional activities (such as an antioxidant activity). The incorporation of the oil in the emulsified form was carried out to avoid the phase separation (exudation of the hydrophobic compound) observed in several works in the literature. The films' mechanical properties and color parameters with emulsified watermelon seed oil were determined. The films were produced using the casting technique, so the viscosity assessment is not a determining factor.

02) As in Figure 1, they have checked Effect on the surface and internal structure of cross-linked cassava starch-based films with different concentrations of watermelon seed oil emulsions (CWSOE, g/100 g of filmogenic solution) but they didn’t check the stability of systems because the selection of emulsion components, is to subject them to external influences such as centrifugal force during the centrifugation process.

Response: Studies related to emulsion stability were carried out in preliminary tests. Film formulations with different oil concentrations (which correspond to the different concentrations of uncooked emulsions) were based on these studies. As defined in preliminary studies, emulsion formulation was described in the work "Watermelon seed oil emulsions were produced according to Dokić et al. [25], with modifications. SA starch (15g) was dispersed in 100 mL of water, and the dispersion was kept at 50°C for 20 min (IKA® C-MAG HS 7 shaker, Staufen, Germany). Subsequently, watermelon seed oil (12g) was incorporated into the starch solution, and the homogenization of the system was carried out at 9500 rpm (20 min, 25 ± 2 °C) using an Ultra-Turrax stirrer (IKA, T25, Germany). (lines 122-127)".

03) A lot of work has been done on watermelon seed oil and emulsions, what is the novelty of your work? How this is different from other research work done previously.

There are studies related to the extraction of watermelon seed oil and emulsions. However, studies involving the PLE technique for oil extraction from watermelon seeds are incipient. Additionally, the incorporation of emulsified oils in biodegradable films aiming at improving the barrier properties is also little explored. The effect of incorporating this oil could still confer antioxidant properties, being of interest to the area. The relevance of the study was highlighted in the conclusions.

04) In conclusion the authors mentioned in the last Although the concentration of phenolic compounds in the films increased as the concentration of WSOE increased, no antioxidant activity was observed. This was probably associated with the composition of the polymeric matrix and the presence of compounds responsible for oxidation.

Probably? It means the authors are not sure about it.

Complementary studies must be carried out. The studies carried out do not allow a conclusion. Some aspects may be associated with these results. The concentration of WSO used, the methodology used to determine the antioxidant activity and the fact that WSO is incorporated in the emulsion form. The presentation of the results is relevant for the area, especially in the case of active packaging.

Reviewer 3 Report

I reviewed the manuscript entitled, Characterization of films produced with cross-linked cassava starch and emulsions of watermelon seed oils extracted with pressurized ethanol. The manuscript is well written with an appropriate introduction section and methodology. Results and discussion are relevant not compared with available scientific literature.

Introduction section is appropriate

Methodology is relevant and cited the references

Figure 1 scale must be improved

Lines 85-86: Other vegetable oils with functional properties are the seed oils of fruits, such as watermelon (OSM)…. What is watermelon (OSM)??

SEM findings must be compared with literature and should improve the discussion

3.2. Atomic Force Microscopy: discussion must be improved with available literature

Figure 3. What is the Y axis ?

Line 420: Duncan test…. It is Duncan’s multiple range test. Please revise throughout the manuscript

Color and antioxidant activity: Please discuss and compare with available literature

What is the control for color?

Line 463: WSOE did not affect …… WSOE showed no affect

Conclusions

Authors should mention the practical application of this film. Any studies conducted? where we can use this film in food packaging, which material can we pack using this film?

References are not according to the journal format. Please revise it

Author Response

We thank you for the opportunity to improve the writing and presentation of the paper entitled “Characterization of films produced with cross-linked cassava starch and emulsions of watermelon seed oils extracted with pressurized ethanol”.

 The comments made by reviewers with their respective corrections are presented below. Most of the reviewers' recommended observations were accepted and completed, including formatting errors and the quality of the images.

 We acknowledge in advance all the cooperation provided by you during the job evaluation process, and we hope that our work will be published in your prestigious journal.

01) Figure 1 scale must be improved

Thank you for commenting. Figure 1 has been improved with higher resolution.

02) Lines 85-86: Other vegetable oils with functional properties are the seed oils of fruits, such as watermelon (OSM)…. What is watermelon (OSM)??

We appreciate the comments.  SM has been changed to the correct nomenclature (WSO- watermelon seed oil).

03) SEM findings must be compared with literature and should improve the discussion

We appreciate the comments. The discussion was improved (Thus, in this context, the application of emulsions in cross-linked starch polymeric matri-ces indicating a good alternative in incorporating hydrophobic compounds. It is possible to verify that the particles remain intact, as also observed by Jiménez-Saelices et al. [16].)

04) Atomic Force Microscopy: discussion must be improved with available literature

The discussion was improved.

Corrections have been made. The text has been modified, and discussions added (“Overall, the incorporation of WSOE increased the roughness of the films, which is a result related to the distribution of emulsions in the film structure. In addition, Tthe roughness is an important parameter, as it affects some properties of the films such as appearance, surface properties and possible interactions with other substances (water, and some food components).

Liu et al. [39] produced films based on Konjac glucomannan with different concen-trations of sunflower oil PE stabilized with cellulose nanofibers and soy protein isolate. Higher roughness were observed in films based on Konjac glucomannan with different concentrations of sunflower oil PE stabilized with cellulose nanofibers and soy protein isolate This study verifiedwere developed, compared  that the films with added PE showed superior roughness to the films without added PE. However, the roughness de-creased when incorporating PE with oil phase concentrations equal to 70% [39].”)

05) Line 420: Duncan test…. It is Duncan’s multiple-range test. Please revise throughout the manuscript.

Thanks for the comments. Corrections have been made.

06) Color and antioxidant activity: Please discuss and compare with available literature What is the control for color?

We appreciate the comments. Calculations and statistical analysis were corrected according to the suggestions.

07) Line 463: WSOE did not affect …… WSOE showed no affect

We appreciate the comments. The correction has been made.

08) Conclusions

The authors should mention the practical application of this film. Any studies conducted? where we can use this film in food packaging, which material can we pack using this film?

Unfortunately, we do not carry out applications. Therefore, we cannot indicate specific products. Studies must be continued and applications evaluated.

09) References are not according to the journal format. Please revise it

We appreciate the corrections. References have been modified. (we used Mendeley).

Round 2

Reviewer 1 Report

Thank you for revising 

Reviewer 2 Report

The authors had  improved the manuscript and answered my question. I recommend the acceptance of the article.